# Perceived Worries and Spirituality: A Mixed Methods Study of the Primary Caregiver Well-Being of Orphan and Vulnerable Children in Ethiopia

**DOI:** 10.3390/children11040380

**Published:** 2024-03-22

**Authors:** Aweke Tadesse, Jesse J. Helton, Kenan Li

**Affiliations:** 1School of Social Work, Saint Louis University, St. Louis, MO 63103, USA; aweke.tadesse@slu.edu (A.T.); jesse.helton@slu.edu (J.J.H.); 2College of Public Health and Social Justice, Saint Louis University, St. Louis, MO 63103, USA

**Keywords:** perceived worry, spirituality, well-being, primary caregiver, Ethiopia

## Abstract

This study investigates the well-being of primary caregivers responsible for orphaned and vulnerable children. Well-being is defined as overall wellness, happiness, and satisfaction. Through mixed methods case studies and purposive sampling, we analyzed data from the Ziway Food for the Hungry Ethiopia program in 2017. Our explanatory analytic approach highlighted issues including resource constraints, chronic illnesses, and community challenges faced by the respondents. Nonetheless, spiritual well-being emerged as a crucial factor for their coping mechanisms. The findings underscore that critical well-being deficiencies require immediate attention. Strategies should prioritize financial and emotional support, emphasizing community capital to enhance the well-being of primary caregivers.

## 1. Introduction

Over 140 million orphans and vulnerable children exist worldwide, with Asia and Africa leading by an estimated 61 and 52 million, respectively [1,2]. The high prevalence of orphans and vulnerable children in Africa is strongly connected with economic and historical determinants such as poverty, natural disasters, war, and other health-related factors such as HIV/AIDS. It was predicted that 55 million people in Africa would die of AIDS between 2000 and 2020 [3,4]. A report from 11 African countries indicated that around 12% of children were orphans, with 11% to 78% having been orphaned due to HIV/AIDS [5]. For over half a century, Ethiopia has been severely affected by extended poverty, war, hunger, HIV/AIDS, and internal conflict and displacement. These factors critically affected the population and contributed to high morbidity, mortality, and social disintegration. Millions of children in Ethiopia have widowed grandparents and relatives who live in distressed communities and struggle with physical, mental, economic, social, and psychological deficiencies [6,7].

Approximately 6% of the Ethiopian population is categorized as orphans and vulnerable children [6]. Similar to other Sub-Saharan Africa (SSA) countries, poverty and HIV/AIDS increasingly affect the well-being of rural households in Ethiopia [8,9]. Stigma and discrimination are widely associated with these two factors and affect orphans and vulnerable children and their primary caregivers mentally, economically, socially, psychologically, and spiritually [10,11].

In Africa, orphans and vulnerable children’s caregivers’ fragile health conditions, limitations in terms of providing basic needs, and lack of adequate care and support are linked to their emotional stress, hopelessness, and perceived fear and worries about their children [1,12,13]. Caregiver well-being deficiencies in several instances, such as mental illness and emotional stress, were associated with children’s low educational achievement [2,14,15]. Most orphans’ and vulnerable children’s primary caregivers in rural Africa, including in Ethiopia, do not receive government support and formal training that could help them cope with these challenges [9,16]. The only exception may be if they have a chance to be included in and qualify for receiving care and support from non-governmental local and international organizations or consistent family support [17,18,19]. Little attention has been given to how vulnerable children’s primary caregivers perceive the future and their coping behaviors in Ethiopian rural settings.

For our purposes, we conceptualized “well-being” not as a single-item measure but rather as a complex construct encompassing a multitude of physical, economic, social, psychological, nutritional, and spiritual components. It also does not mean an absence of sickness or deficiencies in the overall condition of subjective wellness (i.e., in regard to the evaluation of the overall determinants and conditions of quality of life [20]. Well-being has been operationalized as encompassing multiple domains of health, including food and nutrition, general health, mental health, housing, education, spirituality, and community cohesion and safety [21,22]. Operationalizing well-being as consisting of quantitative indicators allows for the measurement of need and the determination of where to focus services. For example, according to Spier’s report, 57% of child caregivers could not provide adequate food; in some regions such as Gambela, 75% of caregivers were receiving food support, 83% visited professional health providers or local health clinics when their children were sick, and 72% lacked basic knowledge on hygiene and sanitation. Other studies indicate that emotional, physical, material, and informational support needs to be improved to enhance Ethiopian orphan and vulnerable children caregivers’ well-being, support their nurturing role, and reduce the risk factors that deteriorate their quality of life and that of their children Other studies also suggested the need for intervention in helping to enhance the coping skills of caregivers raising children under extreme poverty, stigma, worries, and discrimination [23,24].

### 1.1. Ethiopia as a Case Study for Well-Being

A recent study categorized Ethiopia as “fragile” due to various existing humanitarian crises such as war, instability, displacement, hunger, and internal conflict coupled with its existing extreme poverty, economic crisis, and high inflation [25,26,27]. In the last five years (2017–2022), the country has been embroiled in an extended ethnic and political war that has destroyed hundreds of public institutions, including schools, universities, hospitals, clinics, and factories [28]. It has also caused the death of over half a million people and the displacement of millions of people in the North and West regions. As studies have indicated, between the years 2019 and 2020, the Gross Domestic Product (GDP) dropped from 8.36 to 6.06, and the unemployment rate rose from 19% in 2019 to 21.6% in 2020 [29]. For example, the country’s food insecurity accounted for 45% of child deaths. Globally, Ethiopia also has the second-highest population of orphans and vulnerable children [30,31]. Due to poverty, a high unemployment rate, urbanization, migration, internal displacement, ethnic conflicts, and global changes, the traditional support system is becoming fragile and inviting humanitarian organizations to engage in crisis mitigation, relief, and community development interventions [28,31]. However, between 2021 and the beginning of 2023, until the peace agreement between the federal government of Ethiopia and Northern Tigrayan leaders, which was led by the US government and the African Union, the condition of the IDP worsened, coupled with an extreme increase in living standards, and drought affected over 20 million people, a situation currently (i.e., 2023) involving multiple forms of international humanitarian aid for crisis and relief mitigation [26,30].

Ethiopia’s consistent poverty and humanitarian crises have invited many international non-governmental organizations (NGOs) to intervene in development, relief, and crisis emergency responses. Food for the Hungry has been implementing relief and development interventions in Ethiopia since the early 1970s. Regions and populations with critical humanitarian crises were the major operational targets for Food for the Hungry Ethiopia. Food for the Hungry is an international community-based Christian relief and development organization working in over 21 countries worldwide, focusing on the most vulnerable segment of society in low-income countries severely affected by poverty and disasters [32]. As the result of recurring droughts and the high rate of unemployment in Ethiopia’s Rift Valley regions, including the Ziway area, Food for the Hungry Ethiopia initiated a long-term child development intervention in the Ziway area in 2005 [33]. FHE also implemented child development interventions primarily focusing on providing educational, psychological, and spiritual support to orphans and vulnerable children and their caregivers through private sponsorship funding, largely from Americans. FHE implementation models have had ample room to work with three key community partners: community leaders, churches, and families. Local community promoters are the key players and agents in relations between the agency and the target community, households, orphans, and vulnerable children.

### 1.2. The Role of Religiosity: Affiliation and Spirituality

Studies from SSA have shown a link between orphans and vulnerable children caregivers’ well-being and resilient coping strategies and perceived religion and spirituality. For example, a study from Zimbabwe and Ghana indicated that 94% and 84% of orphans and vulnerable children’s caregivers (i.e., grandparents) received support through their church affiliation, respectively [34,35]. Key factors that connect caregivers with the church include survival, staying associated with friends and supporting each other, participating in religious activities, emotional and physical support, and praying and asking God for help and intervention (including miracles) in relation to sustaining them and their orphans and vulnerable children [8,36,37,38]. A study from four different African countries suggested that among six strategies used to sustain caregivers’ mental health, religious practices (i.e., prayers) were the strongest positive indicator across all the countries that 44–67% practiced [39]. The findings also emphasized that participants used spiritual songs to sustain positive emotions.

Meditation and the belief in “doing good” drew in 38% of participants from Kenya and 58% of Cambodians, respectively, and motivated them to connect with other church members and participate in religious activities. Religious orientation and teachings also emphasize promoting social connection and providing care and support for orphans and vulnerable children, poor individuals, and households facing suffering and hardships [37]. Despite the positive role of religion, in contributing to the role and well-being of primary caregivers, information, particularly from studies conducted in Ethiopia among the most vulnerable caregivers of orphans and vulnerable children in semi-urban contexts, is scarce.

### 1.3. Current Study

Knowing that well-being includes a dynamic interaction of multiple areas of health and safety, in this study, we set out to utilize mixed methods to provide a more comprehensive understanding of the construct within the context of distressed communities in Ethiopia. This study aims to explore two specific themes. First, it explores factors related to primary caregivers’ worries that could be directly or indirectly linked to their well-being and nurturing role. Second, it assesses the influence of primary caregivers’ spirituality (religious perceptions and affiliations) in their coping and caring role.

## 2. Materials and Methods

### 2.1. Research Site

Data were collected in Central Ethiopia in the Adami-Tulu Jido Combolcha region and the Ziway area. The study area is in the Great Rift Valley zone, and an estimated population of 49,400 inhabits the Ziway area [33]. This area is mainly characterized by an irregular rainfall distribution and is prone to recurrent drought and critical food shortages [32]. Ziway Lake attracts horticultural investment and daily laborers with low education, including many young women from the southern regions. Ziway town has become home to multi-ethnic groups, which has also caused frequent ethnic conflicts in the last decade [40]. Food for the Hungry Ethiopia has implemented HIV/AIDS care and child development interventions since 2005 [33].

### 2.2. Participants

The study participants were selected from the Food for the Hungry, Ziway HIV, and Child Development projects. Six study cases were chosen from the Ziway area and the villages of Bullbulla and Abossa, and all of those involved participated in the study. The selected households and caregivers were grandparents, single parents, and guardians with no blood relationships. Except for one case (case 2, Zidan), all the participating caregivers were female. The study participants’ ages ranged between 46 and 67 years old. In the selection process, we used the following four criteria aligned with the organization’s initial selection standards: (1) caregivers living with school-age orphaned children and who have worked closely with community workers for at least the last three years, (2) those living in the Ziway area or nearby localities and receiving support from the program for at least for the last three years, (3) those registered in the organization’s official database, and (4) the availability of original child–caregiver case files in the organization’s archive.

#### Researchers

Data were collected by the first author, an Ethiopian who completed his undergraduate education in Ethiopia. He completed his Master of Social Work degree in the USA and is currently a social work Ph.D. candidate. For over 10 years, he led multiple community development intervention programs, including education, nutrition, disaster mitigation, income generation, and sanitation in Mozambique. During that time, he visited hundreds of orphans and vulnerable children’s households and dozens of agencies. The second author is an American with over 20 years of child development research experience, including in qualitative and mixed methods studies. The third author is Chinese and specializes in mobile health and ubiquitous health, focusing on caregiving for children with pediatric asthma. All authors are caregivers of young children and adolescents.

### 2.3. Procedures and Data Collection

We employed a purposive sampling case study method and used criterion sample selection mechanisms [41,42,43]. Data were collected in a sequential pattern. First, a pilot study was conducted, and minor corrections were made to improve the semi-structured interview questions in terms of the study context. Second, structured questionnaires were used to assess the primary caregiver’s well-being domains using a 36-item Likert scale. Third, an in-depth investigation was conducted using semi-structured questions. Scholars have indicated that a case study approach allows for a variety of investigation methods (i.e., both qualitative and quantitative) for collecting data and triangulating findings intriguingly, also giving researchers room for flexibility to draw valid and reliable conclusions [44,45,46]. According to Yin and Creswell, for a real-life and contemporary phenomenon in a natural setting with regard to exploring the “how” and “why” questions in depth, a case study is a preferred approach [43,47,48].

This study utilized a mixed methods approach for data collection. Employing quantitative and qualitative methods in a single study has unique benefits. Among these benefits, data triangulation, complementarity, and expansions are the major ones that could also increase a study’s validity and strengthen its conclusions [41,45,46,49]. This study applied a logic-based thematic analysis to triangulate information from all the source data gathered. Robert Yin also suggested using a logical technique for case study research analysis for integrating observed events and repeated cause–effect meaningful linkages into theoretical prediction [44,50]. Selective coding was used to categorize themes from case interviews and informant group discussions.

### 2.4. Data Analysis and Instrumentation

The current study used a sequential exploratory data analysis, moving from a general survey assessment to a specific in-depth qualitative exploration data analysis approach. First, orphans and vulnerable children’s primary caregivers’ general well-being was assessed using a structured questionnaire. Second, in-depth exploration was carried out using data triangulation from individual interviews, key informant focused group discussions, and individual archival records. A 36-item Likert-type Well-being Measuring Tool (WMT) was used to access individual perceived general well-being domains. This tool is widely used (i.e., in monitoring interventions) to measure well-being domains for vulnerable low-income orphans and vulnerable children contexts in SSA (e.g., Ethiopia, Rwanda, and Uganda) and areas where Catholic Relief Services (CRSs) are operational worldwide. Strong internal consistency (Cronbach’s α up to =0.80) was recorded [21]. Operationally, well-being is viewed as a caregiver’s overall evaluation of quality of life, including a positive sense of overall wellness, happiness, and satisfaction. Three to four questions were used to access each domain, with a total summative score between 1 and 3 for a single domain and 10 and 30 for ten domains, including economy; food and nutrition; shelter; general health; mental health; protection; education; spirituality; family support; and community cohesion. The cutoff points for desirable, average signs of crisis are 25 and above, 23, and 15, respectively. A score below the average indicates deficiencies within some domains, and a score below 15 requires immediate intervention.

In this study, the spirituality domain assessed whether caregivers perceived support as a means of coping with a divine being, “God,” and drew some spiritual support from individuals or their faith community. The WMT included 3-item questions for assessing caregivers’ spirituality linked with their well-being and coping skills. Caregivers were given a chance to respond to three questions and provide a rating from 1 to 3 based on the choices given: “never = 1”, “sometime = 2”, and “all of the time = 3”. These scaled questions were, first, “my belief in God gives me strength to face difficulties”, second, “my belief in God gives me comfort and assurance”, and lastly “my faith community is important to me”. The desired, average, and low scores for a single domain are equal and above 2.5 and 2.3 and below 1.5, respectively. A score below 1.5 indicates a critical crisis condition within a domain. While the WMT spiritual assessment tool examined how caregivers expressed dependability on God, the individual interview aimed to explore spiritual support from the faith community they were affiliated with and their spiritual perceptions. In this study, participants were described by both pseudonyms and case numbers: case-1 to case-6.

## 3. Results

In Table 1, descriptions of the case (participants) characteristics are provided. The caregivers ages ranged between 41 and 67, and except for Zidan (case-2), the rest were females. Geographically, the caregivers included in this study lived within a 15-mile radius in three localities of the Ziway area. Except for Zidan and Zewditu (case-6), the rest had no formal education (illiterate). Meymuna (case-1), Masho (case-3), and Negle (case-5) were uneducated grandparents in their 60s and had a weak religious affiliation, whereas Zeritu (case-4) had no blood relationship (guardian) with the orphans and vulnerable children she cared for, and Zidan and Zewditu were biological parents. They based their livelihoods on selling small goods in the open local market when their health and weather conditions allowed, whereas Masho had a small farming plot that had temporarily been given to family members to use but for which production was to be shared. The family size and number of children under the caregivers’ care ranged from two to eight and one and four, respectively, and Negle and Zewditu had large family sizes (with six and eight members, respectively). Due to caregivers’ low well-being, age, education, and other socioeconomic and demographic conditions, including regarding caring and parental accountability, they had unstable income sources. None of them were able to suggest a range of their monthly income, except the monthly support they were receiving from the agency, which included an in-kind provision of rental support of ETB 300 (USD 13.6), a 1⁄2 ration of beans, 15 KG of wheat flour, and 1⁄2 a liter of cooking oil. However, since the support was only for HIV-infected households and chronically ill caregivers, Zewditu was excluded, even though she and her family suffered from critical well-being deficiencies compared to the other caregivers.

### 3.1. Perceived Well-Being

Table 2 shows that OVC caregivers’ quality of life was significantly affected by the well-being deficiencies in multiple major domains that were theoretically connected to determining their well-being. The findings indicated that while only Zidan and Zeritu had deficiencies in four and five domains, the rest had deficiencies in nine to ten domains. In Table 2, a summary of the caregiver’s well-being domain scores (range from 10–30) is presented, and none of the cases show desirable (25) or average (23) scores; instead, deficiencies can be observed (a score below 22), except for case 2 (22.1). Zewditu indicated deficiencies that require immediate crisis intervention (a score below 15).

In agreement with the findings from the survey, the exploration of the key-informant focus group discussion confirmed that all of the participants agreed that these caregivers were stressed and that they and their orphans and vulnerable children faced huge burdens of their own. For example, the para-social workers said, “The caregivers themselves don’t have anything. As community workers, we all know what we have been experiencing whenever we visit their homes…we cry [with them]and then share our salaries. Caregivers do not have a moment to go without thinking about what to eat, their shelter, health, and clothes. They always tend to over-depend on us and shift their responsibilities to us”. C-4 said, “We were five in this one room [4 × 3 muddy house], and two of my grandchildren left because I could not feed them. Moreover, people around here hate us; we are afraid to go out, so I stay home. I heard that people with HIV are getting better and hope to live longer”. Zewditu said, “… because these baby twins, I can not work anymore; they just began eating food. I am very bitter this year. People said bad words about my twins and told me they were not surviving. Whenever they feel unhealthy, I am stressed. This is a hard place to live, and people are harsh and sympathetic. People know I am a stranger, and no one is visiting us except Food for the Hungry workers”. Except for Zidan, all the primary caregivers felt isolated, less accepted, and discriminated against by the community they were living in.

### 3.2. Spirituality among Orphans and Vulnerable Children with respect to Caregiving Roles and Well-Being Conditions

Figure 1 indicates that compared to all other domains, with an average score of 2.3, the spirituality domain had the highest mean score for C-1 (2.5), C-3 (2.3), C-5 (2.3), and C-6 (1.6), and it is the only domain that all caregivers reported to be above 1.5, with no indication of critical deficiencies (crises) that required immediate action.

The data from the individual interviews indicated that the primary caregivers’ spirituality was strong. During the discussion, they repeatedly mentioned “God” while describing their burdens, joy, health, and future perceptions. For example, Meymuna (C-1), a 67-year-old grandmother and Muslim raising three children who lost both parents because of AIDS, said, “God gave me all, poverty, grief, and illness altogether. I do not want to complain; He is comforting me through these boys; they are happy and good students”. Zeritu (C-4), a 55-year-old HIV-infected protestant grandmother raising three orphans and vulnerable children, two of whom were abandoned by their mothers, said, “People usually say children are a blessing and medicines. Still, for me, they are the source of endless suffering. My daughter left me with two children; she disappeared to the Middle East long ago. There was a time when I thought bad things about my life and my daughters’ abandonment. But spiritual advice breaks a person’s heart”. Similarly, Zidan, a 47-year sick father of an 11-year-old boy and a follower of Islam, said, “When my wife died, I was in huge trouble, and I do not know what and I do not have anything to feed my son properly. God provided my neighbors who are taking care of my son. People say God didn’t quarrel and discarded us without celebrating with us first (i.e., culturally means, in the middle of suffering and darkness, God provides a way out)”. Even though all the caregivers vividly expressed their dependability on God, to the contrary, except for C-2, all the rest expressed that they were not connected with their faith community or receiving any support.

### 3.3. Orphans and Vulnerable Children PCG’s Perceived Worries

Semi-structured questions were used to probe and explore factors and feelings connected to the primary caregivers’ perceived worries. These questions included, “What are your worries for your future life if you have”; “What are the issues that make you constantly worry about?”; and “Can you share how or why the issue makes you worry”? In addition, a sense of worry was explained. The same perception was related to the feeling of being burdened by their basic needs, health conditions, their children’s future, community cohesion, their children’s education, and the continuity of support from the agency. For some of the caregivers, the grief and brokenness stemming from the loss of their beloved ones (children and partners), particularly in relation to AIDS, were still on their minds and created hopelessness and fear for the future of their own and their children. Meymuna, a 67-year-old sick grandmother of three orphans and vulnerable children, said,
“People in this area were admiring the life and beauty of my daughter; she was like a flower but cut short when she got infected with HIV. When I see her picture now, I turn away and cry; I wish I had died before”. I am old, weak, and a cancer patient.”. After this, she was deeply emotional and cried loudly! She continued, “I always worry about these children; I do not know what the future holds for these little flowers. I worry that they might run into trouble and be left homeless. I have no property to pass to them, neither land nor a house. I do not have good friends because my husband was jealous. People around these are friendly towards us; because of our poverty, my daughter died of HIV. When poverty becomes your enemy, everything around you turn against you, and you remain like a person necked, ashamed, and powerless. But in the middle of all these, no one knows what God can do. Life is colorful”.

A common and strong sense of worry and hopelessness among the primary caregivers was linked with the information they were told, i.e., that the agency would phase out from the area and that the support they had been receiving for a long time (four to eight years) would cease in a year or two. They had received a monthly handout of 1⁄2 KG of beans, 15 KG of wheat flour, and 1/2 L of cooking oil and ETB 300 (USD 13.6) of rental support. Besides this food aid, the agency provides educational materials and school uniforms for each school-attending child. However, all the caregivers were deeply worried about their children’s future survival; none showed a sense of self-reliance in shaping their orphans’ and vulnerable children’s futures.

## 4. Discussion

The context of prolonged poverty and recurring humanitarian crises in Ethiopia is related to an increase in families with orphans and vulnerable children (over 5.5 million) and low socioeconomic status (SES), which, in turn, is connected to well-being deficiencies, increased child mortality and social disintegration, and a low GDP [44,50]. Orphans’ and vulnerable children’s primary caregivers in Ethiopia have critical resource and health limitations with respect to addressing basic needs and fulfilling proper caring and nurturing roles. Besides historical determinants such as poverty and HIV-AIDS, the prolonged socioeconomic fragility and recurring humanitarian crisis in this country are also factors influencing the high number of orphans and vulnerable children and the vulnerability of their caregivers. These factors have also left millions of orphans and vulnerable children without biological parents and under the care of aged and disabled grandparents with limited development options [25,26,27].

The findings of this study are consistent with those of other studies worldwide, particularly with regard to low-income regions in SSA, where orphans’ and vulnerable children’s primary caregivers (i.e., disabled and aged grandmothers, HIV/AIDS-infected patients, and single parents with no job but with large households) live with multiple well-being deficiencies of their own and struggle to address the basic needs of their orphans and vulnerable children [7,28]. The results show that even though PCGs were receiving institutional care and support, their perceived well-being was below average, which indicated deficiencies indicating a crisis that may require immediate action for intervention. Further, this exploration revealed that caregivers live with perceived worries, fear, and hopelessness for their own and their children’s future survival, safety, security, and education.

Perceived stress and burdens were linked with the caregivers’ fragile socioeconomic and health conditions, including a lack of a supportive community and assets such as land and houses that could be passed down to children. Studies have suggested that, in homogeneous settings and relational community settings, neighborhood connections have a positive association with well-being factors [51,52,53,54], but vulnerability (i.e., an extreme poverty status) and HIV could have a negative effect [8,9]. Similarly, in this study, all cases (households) with HIV/AIDS patients felt isolated and unwelcome. However, in cases with no HIV issues (e.g., case 2), neighborhood connection was beneficial and impactful in providing care and support for orphans, vulnerable children, and single fathers. On the other hand, the image painted of “HIV-AIDS” in the community could be a factor that prevented primary caregivers with HIV cases from approaching others, including their faith community, and seeking help.

In the local setting, concerning churches, the respondents did not indicate that a local church had paid attention to or tried to reach and assist the HIV-affected households. However, the caregivers’ spirituality and belief in God were still used as a coping strategy that could help regulate their perceived sources of worries and hopelessness. In agreement with the previous studies, the caregivers’ spirituality played a coping role in providing psychological care, reinforcing a positive feeling and perception despite their survival struggles and hardships, which relates to trust in a divine being, “God”, who knows their situation and controls the future [8,38,39]. In this study, the caregiver’s spirituality well-being domain indicated that the score was higher than the other nine domains included in the WMT scale.

Despite its contribution to the evidence regarding OVC PCGs’ well-being conditions, worries, and the role of spirituality, this study has limitations. First, the findings are based on a case study exploration and thus do not allow generalizability or the establishment of causality. Second, a few cases were selected based on criterion purposive sampling, which does not represent other cases in the program (i.e., lacking random selection). Third, the cases were selected only from the Ziway area. However, the findings could contribute to a further comprehensive investigation, intervention science, and practices that address well-being issues in a low-socioeconomic-status and poverty-stricken environment.

## 5. Conclusions

Orphans’ and vulnerable children’s primary caregiver’s vulnerability is complex and deep. Nonetheless, formal humanitarian care and support are critically needed to support the short- and long-term survival, not just well-being, of caregivers and their children. Our results show that part of that support could include integrating adaptable and sustainable approaches outside of economic and nutritional support. Indeed, focusing on spiritual coping could help lower perceived worries, hopelessness, and fear of long-term survival for families like those in our study if their basic nutritional, health, and economic needs are also supported. For our participants, a low perception of family support and community cohesion was a source of worries and insecurity. Therefore, it may be helpful to monitor sources of worries not connected to economic or nutritional health, such as feelings of insecurity, isolation, loneliness, and characteristics of community cohesion, including relationships with their faith communities. Regardless of their practical implications, our results show that further rigorous and comprehensive scientific research is needed to examine a more comprehensive set of well-being determinants of orphans and vulnerable children and their caregivers in Ethiopia, including the role of spiritual coping.

## Figures and Tables

**Figure 1 children-11-00380-f001:**
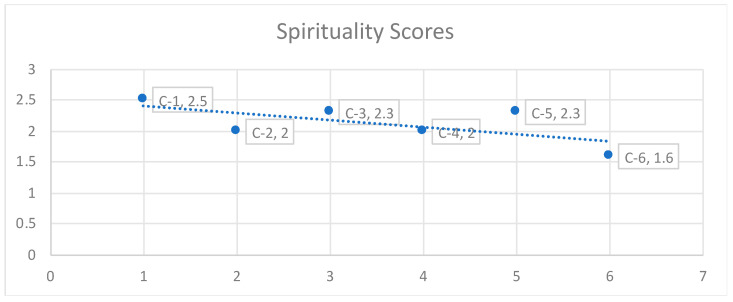
Orphans’ and vulnerable children’s primary caregivers’ scores for spirituality domain.

**Table 1 children-11-00380-t001:** Participants demographic characteristics.

Case Code	Pseudonyms	Age	Gender	Relationship to the Orphan and Vulnerable Child	Means of Living
C-1	Meymuna	67	Female	Grandmother	Selling cooking charcoal
C-2	Zidan	49	Male	Biological Father	Daily laborer
C-3	Masho	61	Female	Grandmother	Selling vegetables
C-4	Zeritu	55	Female	Guardian	No work (mostly sick)
C-5	Negele	66	Female	Grandmother	Small farming
C-6	Zewditu	46	Female	Biological mother	Sale of house-made goods

**Table 2 children-11-00380-t002:** Summary of PCGs’ well-being domain mean scores.

Case Code	Well-Being Domain Scores	Score Descriptions
C-1	16.2	Below the average and deficiencies within nine domains
C-2	22.1	Below the average and deficiencies within four domains
C-3	17.95	Below the average and deficiencies within nine domains
C-4	20.7	Below the average and deficiencies within five domains
C-5	16.55	Below the average and deficiencies within nine domains
C-6	14.5	Below the average and deficiencies within nine domains

## Data Availability

The data presented in this study are available on request from the corresponding author. The data are not publicly available due to privacy concerns.

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
