# Peer review of "Perceived Worries and Spirituality: A Mixed Methods Study of the Primary Caregiver Well-Being of Orphan and Vulnerable Children in Ethiopia"

_children, 2024, doi:10.3390/children11040380_

Round 1
Reviewer 1 Report
Comments and Suggestions for Authors
The topic is important but there are significant points in the manuscript that need serious attention. Overall, the article is not well structured, some sub-headings are unnecessary and there is a lot of redundancy. In my opinion, the manuscript needs some reconstructed. My other comments are as follows:
​
Abstract:
"Notably Case-6, exhibited the lowest well-being score" This should not be mentioned here.
There is no information about the participants
Introduction:
What does SSA mean? Better to explain the abbreviations first
I think that either the sub-headings should be removed or the factors should be listed first and then elaborated.
126 to 130: Please a citation
Current study
Question 2: Because the studies by Biru et al., 2015; Selamu et al., 2014 have already shown this relationship, I believe this can be changed to a hypothesis
Materials and Methods:
I think the authors can explain this information without the use of subheadings
2-1 Research site: some of this can be moved to the discussion section.
2-2 Mention the age and gender of the participants
Data analysis
Lines 227 to 237 should be moved to Materials and Methods and given "Instrument" title.
Maybe a table with age, gender and related information (such as illness or...) would be useful.
Results
3-1: Demographics, this subtitle is unnecessary.
253-258: Please mention age of the cases
396 to 400: Please revise and reformulate
I also think the manuscript needs English editing
Comments on the Quality of English LanguageThe manuscript needs a thorough English editing.
Author Response
“The topic is important but there are significant points in the manuscript that need serious attention. Overall, the article is not well structured, some sub-headings are unnecessary and there is a lot of redundancy. In my opinion, the manuscript needs some reconstructed. My other comments are as follows:
​
Abstract:
"Notably Case-6, exhibited the lowest well-being score" This should not be mentioned here.
There is no information about the participants
Introduction:
What does SSA mean? Better to explain the abbreviations first
I think that either the sub-headings should be removed or the factors should be listed first and then elaborated.
126 to 130: Please a citation
Current study
Question 2: Because the studies by Biru et al., 2015; Selamu et al., 2014 have already shown this relationship, I believe this can be changed to a hypothesis
Materials and Methods:
I think the authors can explain this information without the use of subheadings
2-1 Research site: some of this can be moved to the discussion section.
2-2 Mention the age and gender of the participants
Data analysis
Lines 227 to 237 should be moved to Materials and Methods and given "Instrument" title.
Maybe a table with age, gender and related information (such as illness or...) would be useful.
Results
3-1: Demographics, this subtitle is unnecessary.
253-258: Please mention age of the cases
396 to 400: Please revise and reformulate
I also think the manuscript needs English editing”
Our Responses:
Dear Reviewer,
Thank you for your insightful comments and suggestions regarding our manuscript. We appreciate the time you took to review our work and provide constructive feedback. Below is our point-by-point response to your comments:
- Original Comment: "Notably Case-6, exhibited the lowest well-being score" should not be mentioned here.
- Response: We agree with your observation and have removed the specific mention of "Notably Case-6 exhibits the lowest” from the abstract and replaced it with “ The critical well-being defects”. We have revised our abstract to keep it concise and focused on the general findings and significance of the study.
- Original Comment: What does SSA mean? Better to explain the abbreviations first.
- Response: We apologize for the oversight. The abbreviation 'SSA' stands for Sub-Saharan Africa. We have now included clarifications at the first instance of all standard abbreviations used in the text and removed all the non-standard abbreviations.
- Original Comment: Suggestion to either remove sub-headings or list factors first and then elaborate. Also, a citation is needed for lines 126 to 130.
- Response: We have revised the structure of the introduction for better flow and clarity. The factors are now listed first, followed by a detailed elaboration. Additionally, for the description in lines 126 to 130, we made a connection with a new citation [Preschool-Bell et al., 2019]”
- Original Comment: Change the question to a hypothesis due to existing studies.
- Response: We appreciate this suggestion. The mentioned question has been reformulated into a hypothesis, acknowledging the findings of Biru et al., 2015, and Selamu et al., 2014.
Changes: Line 190 – Line 194 in the revised document.
- Original Comment: Suggestion to explain information without the use of subheadings.
- Response: We have revised this section for better continuity and flow, minimizing the use of subheadings while ensuring clarity of information.
Changes:
- Subheadings of “socioeconomic, geography and organizational interventions” were put under a single sub-heading “Socioeconomic Condition”.
- The additional structural change was also made under comment # 7 “ sub-topics “Procedure & Data collection” merged” and sub-heading “Analysis” revised “ to “Analysis & Instrumentation”
- Original Comment: Move some information to the discussion section and mention age and gender of participants.
- Response: The age and gender of the participants are now clearly stated in the Materials and Methods section.
Changes: Line 211 – Line 212.
- Original Comment: Move lines 227 to 237 to Materials and Methods and title it "Instrument".
- Response: As suggested, we have moved these lines to the Materials and Methods section under the new subheading "Instrument." This change provides a clearer context for the data analysis methods used in the study.
Changes: “Procedures and Data collection” combined, and lines 227 to 237 are placed under the sub-heading “Analysis & Instrumentation”.
- Original Comment: A table with age, gender, and related information would be useful.
- Response: We have included a new table (Table 1) presenting the demographic information (age, gender, etc.) of the participants, along with other relevant details. This table aims to provide a quick reference for readers.
Change: Table 1 was added to present “Participant Demographic Characteristics”
Case Code |
Pseudonyms |
|
|
Age |
Gender |
Relationship to the Orphan and Vulnerable Child |
Means of Living |
C-1 |
Meymuna |
|
|
67 |
Female |
Grandmother |
Selling cooking charcoal |
C-2 |
Zidan |
|
|
49 |
Male |
Biological Father |
Daily laborer |
C-3 |
Masho |
|
|
61 |
Female |
Grandmother |
Selling vegetables |
C-4 |
Zeritu |
|
|
55 |
Female |
Guardian |
No work, mostly sick |
C-5 |
Negele |
|
|
66 |
Female |
Grandmother |
Small farming |
C-6 |
Zewditu |
|
|
46 |
Female |
Biological mother |
House-made |
- Original Comment: Remove unnecessary subtitle "Demographics" and mention the age of the cases.
- Response: The subtitle "Demographics" has been removed to streamline the section. The ages, pseudonyms, gender, relation to the orphans and vulnerable child, and means of livelihood of the cases are now explicitly mentioned in the text, providing a clearer picture of the study population in the newly added Table 1.
- Original Comment: Revise and reformulate lines 396 to 400.
- Response: These lines have been thoroughly revised for clarity and coherence.
Change: Lines 443 - 446 in revised version.
- Original Comment: The manuscript needs English editing.
- Response: We have conducted a comprehensive English language edit to improve the readability and academic tone of the manuscript.
Changes:
For example, we checked the standard and non-standard abbreviations and made appropriate revisions. We made a significant revision (re-written the subtitle “ The research site, lines 149-164” and replaced it with the following concise description:
“Data was collected in Central Ethiopia, the Adami-Tulu Jido Combolcha region, and the Ziway area. It is in the Great Rift Valley zone, and an estimated population of 49400 inhabited the Ziway area (FHE, 2009). The area is mainly characterized by irregular rainfall distribution and is prone to recurrent drought and critical food shortage (FHI, 2006). Ziway Lake attracted horticulture investment and daily laborers with low education, including many young women from the southern regions. Ziway town has become home to multi-ethnic groups, which has also caused frequent ethnic conflicts in the last decade (Schewel, 2018). Food for the Hungry Ethiopia has implemented HIV-AIDS care and Child Development interventions since 2005 (FHE, 2009).
Similarly, we made a significant revision to the subtitle “participant site, line 193-205” and replaced it with the following description:
“The study participants were selected from the Food for the Hungry, Ziway HIV and Child Development project. Six study cases were chosen from the Ziway area, Bullbulla, and Abossa villages. The selected households and caregivers were grandparents, single parents, and guardians with no blood relationships. The selection used the following four criteria aligned with the organization's initial selection standards. These are 1) Caregivers living with school-age orphaned children and closely working with the community workers for at least the last three years, 2) Living in Ziway areas or nearby localities and receiving support from the program for at least for the last three years, 3) Registered in the organization official database, and 4) Availability of original Child-caregiver case file in the organization archive.”
We hope that these revisions and responses adequately address your concerns. We are committed to enhancing the quality of our manuscript and appreciate your guidance in this process.
Sincerely,
All authors
Reviewer 2 Report
Comments and Suggestions for Authors
I have reviewed the manuscript, titled “Perceived Worries and Spirituality: A Mixed Methods Study of Primary Caregiver Well-being of Orphan & Vulnerable Children in Ethiopia”. The aim was to examine, what factors relate to primary caregivers’ worries (hope) that could be directly or indirectly linked to their well-being and role as nurturers and whether orphans and vulnerable children primary caregivers' spirituality (religious perception and affiliation) is linked with their resilience/coping and caring role.
The study has some strong points (clear introduction, an interesting sample, constructive discussion).
However, I would like to ask the authors to address some points in order to improve the paper.
Introduction:
1) The Introduction section should contain more information on well-being and resilience (underlying mechanisms).
2) Can you relate to any theory or model that could explain associations between worries and well-being? The current version of the introduction clearly lacks a theoretical approach, which needs to be described.
3) There should be a clear distinction between resilience/coping as they are two different factors in psychology.
Method:
4) Did any of the invited participants not volunteer to take part in the study?
5) The section 2.2. Participants should contain more information.
Discussion:
6) Can you elaborate on the following statement: “The result showed that even though PCGs were receiving institutional care and support, their perceived well-being was below average, which indicated defects with a signal of crisis that may need immediate action for intervention.”(page 9).
7) P. 10: The statement: “the caregiver's spirituality and belief in God were still used as a coping strategy that could help regulate their perceived source of worries and hopelessness”. Can you specify your evidence?
Author Response
Comments from Reviewer 2:
I have reviewed the manuscript, titled “Perceived Worries and Spirituality: A Mixed Methods Study of Primary Caregiver Well-being of Orphan & Vulnerable Children in Ethiopia”. The aim was to examine, what factors relate to primary caregivers’ worries (hope) that could be directly or indirectly linked to their well-being and role as nurturers and whether orphans and vulnerable children primary caregivers' spirituality (religious perception and affiliation) is linked with their resilience/coping and caring role.
The study has some strong points (clear introduction, an interesting sample, constructive discussion).
However, I would like to ask the authors to address some points in order to improve the paper.
Introduction:
1) The Introduction section should contain more information on well-being and resilience (underlying mechanisms).
2) Can you relate to any theory or model that could explain associations between worries and well-being? The current version of the introduction clearly lacks a theoretical approach, which needs to be described.
3) There should be a clear distinction between resilience/coping as they are two different factors in psychology.
Method:
4) Did any of the invited participants not volunteer to take part in the study?
5) The section 2.2. Participants should contain more information.
Discussion:
6) Can you elaborate on the following statement: “The result showed that even though PCGs were receiving institutional care and support, their perceived well-being was below average, which indicated defects with a signal of crisis that may need immediate action for intervention.”(page 9).
7) P. 10: The statement: “the caregiver's spirituality and belief in God were still used as a coping strategy that could help regulate their perceived source of worries and hopelessness”. Can you specify your evidence?
Our Responses:
Dear Reviewer,
Thank you for your thorough review and valuable feedback on our manuscript. We are grateful for your recognition of the strengths of our study, including the clear introduction, interesting sample, and constructive discussion. We have addressed each of the points you raised to improve the quality and clarity of our paper.
- Original Comment: The Introduction section should contain more information on well-being and resilience (underlying mechanisms).
- Response: We have expanded the Introduction section to include a more comprehensive discussion on the concepts of well-being and resilience, specifically focusing on the underlying psychological and social mechanisms that contribute to these constructs.
Changes: the following additional description was added:
According to the UNICEF holistic development guidelines, holistic development in an Ethiopia setting includes five key well-being domains: adequate nutrition, good health, early opportunity for learning, responsive caregiving role, and safety and security (Spier et al., 2023; UNICEF, 2014). Nonetheless, there is the existing critical poverty condition, loss of policy enforcement, and lack of support for both the holistic development of orphans and the well-being of vulnerable child caregivers remain a growing community and country problem. For example, according to Spier's report, 57% of child caregivers could not provide adequate food; in some regions such as Gambela, 75% of caregivers were receiving food support, 83% visited professional health providers or local health clinics when their child was sick, 72% lack basic knowledge on hygiene & sanitation. Research indicated that the need for emotional, physical, material, and informational needs to improve orphan and vulnerable children caregivers’ well-being, support their nurturing role, and reduce the risk factors that deteriorate their and their children's quality of life in Ethiopia (Jansen-van Vuuren et al., 2023). This suggestion also considers the growing poverty, hunger, resource limitation, and unstable humanitarian conditions in different parts of the country. Other studies also suggested the need for intervention in assisting to enhance the coping skills of caregivers raising children under extreme poverty, stigma, worries, and discrimination, which relate to a lack of support in empowerment, coping skills, and health, resources, and ethnic related stigma and discrimination (Szlamka et al., 2023; Sisay, 2023).
- Original Comment: Lack of a theoretical approach to explain associations between worries and well-being.
- Response: We have incorporated a theoretical perspective to explain the associations between worries and well-being. This includes references to established models and theories that elucidate these relationships.
Changes: This theoretical section was added in the introductory section:
“An established theory and empirical support indicate that stressful living patterns and low socioeconomic conditions could affect well-being and coping skills and have adverse short-term and long-term effects. For example, the stress theory (Lazarus, 1993) underlines the mediation effect of resources between a stressful environment and an individual's well-being, which also involves cognitive and coping skills (Lever, 2008). Similarly, other scholars also linked stress with individual personality, the nature of daily life practices, and events in a given environment (McCrae & Costa,1986; Vollrath, 2001; Dohrenwend & Dohrenwend, 1969. Scholars also suggested a theoretical and empirical link between the quality of well-being at a later age and socioeconomic conditions at an early stage, which include individuals' financial & health conditions and social skills and behaviors (Moody, 2020; Moen, 2001). For example, the fundamental cause theory (grounded in Lieberson’s “basic cause” concept [19195]) underlines the SES's role in predicting quality of life and health through the positive effect of the availability of flexible resources, knowledge, and skills in social determinants (Phelan et al., 2010).”
- Original Comment: A clear distinction is needed between resilience and coping.
- Response: We have revised the document and used “coping” consistently for avoiding the confusion the word “resilience”.
- Original Comment: Clarification is needed on whether any invited participants did not volunteer to participate.
- Response: We have added information regarding the participation rate of the invited participants, and the participation rate was 100%.
- Original Comment: The section 2.2. Participants should contain more information.
- Response: We have enriched the "Participants" section with additional details and made a significant revision. A clear revised description was provided “
Changes: Section 2.2 was redrafted as the following paragraph:
The study participants were selected from the Food for the Hungry, Ziway HIV and Child Development project. Six study cases were chosen from the Ziway area, Bullbulla, and Abossa villages. The selected households and caregivers were grandparents, single parents, and guardians with no blood relationships. The selection used the following four criteria aligned with the organization's initial selection standards. These are 1) Caregivers living with school-age orphaned children and closely working with the community workers for at least the last three years, 2) Living in Ziway areas or nearby localities and receiving support from the program for at least for the last three years, 3) Registered in the organization official database, and 4) Availability of original Child-caregiver case file in the organization archive.
We have also provided more information on the participant demographic characteristics in the newly added Table 1.
Table 1 Participant Demographic Characteristics
Case Code |
Pseudonyms |
|
|
Age |
Gender |
Relationship to the Orphan and Vulnerable Child |
Means of Living |
C-1 |
Meymuna |
|
|
67 |
Female |
Grandmother |
Selling cooking charcoal |
C-2 |
Zidan |
|
|
49 |
Male |
Biological Father |
Daily laborer |
C-3 |
Masho |
|
|
61 |
Female |
Grandmother |
Selling vegetables |
C-4 |
Zeritu |
|
|
55 |
Female |
Guardian |
No work, mostly sick |
C-5 |
Negele |
|
|
66 |
Female |
Grandmother |
Small farming |
C-6 |
Zewditu |
|
|
46 |
Female |
Biological mother |
House-made |
- Original Comment: Elaboration is needed on the statement about PCGs’ perceived well-being.
- Response: We have elaborated on this statement under the introduction; an additional two paragraphs (around 400 words) were added to elaborate the concept in the study context (the determinants and connected theoretical concepts). Additional information is also given under the “instrumentation: well-being” domains.
- Original Comment: Clarification needed on the evidence supporting the role of caregiver’s spirituality in coping.
- Response: We have provided specific evidence and examples from our data to substantiate the statement regarding the role of spirituality and belief in God as a coping strategy among caregivers. This includes qualitative insights from interviews and quantitative data analysis results.
We hope these revisions adequately address your concerns and contribute to the enhancement of our manuscript. We appreciate the opportunity to improve our work based on your insightful feedback.
Sincerely,
All authors
Round 2
Reviewer 1 Report
Comments and Suggestions for Authors
The authors have addressed all my concerns and I see no reason why the paper cannot be published in its present form.
Author Response
We sincerely appreciate your positive feedback and are delighted to hear that you find our revised manuscript suitable for publication in its current form. We are grateful for the time and effort you dedicated to reviewing our manuscript. It is encouraging to know that our efforts to address the concerns raised have been recognized.